

# The measurement of atmospheric $CO_2$ at KMA/GAW regional stations, the characteristics, and comparisons with other East Asian sites

Haeyoung Lee [1,2], Sang-Ok Han[1], Sang-Boom Ryoo[1], Jeong-soon Lee[3], and Gang-woong Lee[2]

1 Environmental Meteorology Research Division, National Institute of Meteorological Sciences, Jeju, 63568, Republic of Korea
Atmospheric Chemistry Laboratory, Hankuk University of Foreign Studies, Gyeonggi-do, 17035, Republic of Korea
3.Korea Research Institute of Standards and Science, Daejeon, 34113, Republic of Korea

*Correspondence to Haeyoung Lee (leehy80@korea.kr)*

*Abstract. To understand the carbon cycle at policy-relevant spatial scales, a high density of high-quality $CO_2$ measurement sites is needed. In 2012, the Korea Meteorological Administration (KMA) installed $CO_2$ monitoring systems at Anmyeondo (AMY) in west, Jeju Gosan Suwolbong (JGS) in south-west and Ulleungdo (ULD) in east parts of Korea. Three stations were instrumented with identical greenhouse gas measurement systems based on Cavity Ring Down Spectroscopy (CRDS) and a new drying system developed by KMA and Korea Research Institute of Standards and Science (KRISS). This drying system is suitable in the humid areas; water vapour measured by the CRDS in ambient air was 0.001 to 0.004% across the stations. Measurement uncertainties expressed by the quadrature sum of the uncertainties from the drying system, scale propagations, repeatability, and reproducibility were ~ 0.11 ppm from all KMA stations in 68% confidence interval. Average monthly $CO_2$ enhancements above the local background at each station were 4.3 ± 3.3 ppm at AMY, 1.7 ± 1.3 ppm at JGS and 1 ± 1.9 ppm ($1\sigma$) at ULD respectively during 2012 to 2016. At AMY station, located between China and Korea, $CO_2$ annual means and seasonal variations are also greater than the other KMA stations indicating that it is affected not only by local vegetation, but also added anthropogenic sources. Selected baseline $CO_2$ at AMY and at JGS in the west part of Korea are more sensitive to East Asia according to wind direction and speed. Through the comparison of long-term trends and growth rates at AMY with other East Asian stations over 15 years, it was suggested that they could be affected by not only local vegetation but also measurement quality.*

## 1. Introduction

Carbon dioxide, the most important anthropogenic greenhouse gas, is one of the main drivers of climate change on Earth. Measurements of atmospheric $CO_2$ have assumed increased importance to track the increase in global $CO_2$ due to fossil fuel combustion (Canadell et al., 2007; Knorr, 2009).

Roughly half of anthropogenic $CO_2$ emitted by fossil fuel combustion is stored in the atmosphere, whereas the other half is absorbed by the oceans and terrestrial ecosystems. Recent studies showed the atmospheric $CO_2$ network is not yet dense enough to confirm or invalidate the increased global carbon uptake, estimated from ocean measurement or ocean models (Wanninkhof et al., 2012) but emphasized that the combination of a highly dense observation network, coupled with atmospheric models, leads to help understand regional carbon fluxes (Dolman et al., 2009). Therefore, confidence in our understanding of carbon cycle processes may be improved by a higher density of continuous measurement sites.

There are now over 400 regional stations monitoring atmospheric $CO_2$ under the Global Atmosphere Watch Programme (GAW) of the World Meteorological Organization (WMO) (https://gawsis.meteoswiss.ch). These sites capture more regional scale





information on fluxes than global stations, which reflect only well-mixed air mass. However, if technical measurement skill and data quality control are not sufficient, the data may not be useful to help identify and understand changes to the carbon cycle caused by climate change. Also, both measurement uncertainty and imperfect knowledge of the composition of background air can limit the precision of observation-based estimates of local or regional scale greenhouse gas enhancements (Graven et al.,

2012; Turnbull et al., 2009, 2015).

Korea is important due to its location, where it is affected by flow from East Asia, especially China. Korea's atmospheric $CO_2$ monitoring history started at Tae-Ahn Peninsula (TAP, 36°44'N, 126°08'E, 20 m above sea level), in the west part of Korea, in 1990 with weekly flask-air samples as a part of the NOAA/CMDL/GMD Cooperative Global Air Sampling Network (http://www.esrl.noaa.gov/gmd/ccgg/flask.php). Studies demonstrated its regional characteristically high $CO_2$ was affected by

China's industrial regions, while for $CH_4$ it was affected by Russian wetlands and local rice cultivation near TAP (Dlugokencky et al., 1993; Kim et al., 2014).

Since 1999, the Korea Meteorological Administration (KMA) has been monitoring atmospheric $CO_2$ at Anmyeondo (AMY, 36.53°N, 126.32°E, 46 m above sea level from a 40 m tower), about 28 km from TAP. Nevertheless, this was the first attempt to continuously monitor $CO_2$ in Korea. In 2012, KMA expanded its monitoring network to include data from the south-west (Jeju

Gosan Suwolbong, JGS, 33.30°N, 126.16°E) and the east (Ulleungdo, ULD, 37.48°N, 130.90°E) parts of Korea to cover the whole peninsula for a better understanding of $CO_2$ sources and sinks. At the same time, all three monitoring stations started to use analyzers based on Cavity Ring Down Spectroscopy (CRDS; a different model at each station, Picarro, CA, USA) with the same measurement method. So far, even though measurements began in 1999 at AMY, there is no published description of methods used to measure and process the data from the three KMA sites.

In this paper, we present the measurement and sampling system, data quality and processing methods, and analyze the characteristics of $CO_2$ at those three KMA monitoring stations during 2012 to 2016. We calculate measurement uncertainties separately from the hourly, daily and monthly standard deviations, which include natural variability and measurement uncertainty. We compare the data to other stations in East Asia: the global background WMO/GAW station in Waliguan (WLG, 36.28°N, 100.90°E, 3810 m), China, and Ryori (RYO, 30.03°N, 141.82°E, 260 m), which reflects global growth rates as a

regional WMO/GAW station in Japan (Watanabe et al., 2000). In addition, we present 15 years of long-term $CO_2$ measurements in East Asia, including those from AMY. Furthermore, this paper will serve a reference for KMA data archived at the World Data Centre for Greenhouse Gases.

## 2.      Experiment

### 2.1      Sampling sites

The locations of Anmyeondo (AMY), Jejudo Gosan Suwolbong (JGS), and Ulleungdo (ULD) stations are shown in Fig. 1 with their details summarized in Table 1.

AMY is located in the west part of Korea, about 130 km southwest from the megacity of Seoul. Within a 100 km radius, the semiconductor industry and relevant industries exist. Also, the largest thermal power plants fired by coal and heavy oil in Korea are within 35 km to the north-east and south-east of the station. Local activity is related to agriculture, such as rice paddies, sweet

potatoes and onions, while the area is also known for its leisure opportunities during summer. The west and south side of AMY is open to the sea and along the coast, there is a large tidal mudflat with many pine trees.





JGS is located in the west part of Jeju Island, which is the biggest volcanic island (1,845.88 km$^2$) in the south-west of Korea and about 90 km from the mainland. Jeju is popular for tourists regardless of the season, while the region of Suwolbong is famous as a Global Geo-park due to the outcrops of volcanic deposits exposed along the coastal cliff where JGS is located. Next to JGS there is a wide plain where mainly potatoes, garlic and onions are harvested. The side of the station from south-west to north-

west is open to the sea, where there are volcanic basalt rocks. The sea to the south is connected to the East China Sea and the sea to the west is linked to the Yellow Sea.

ULD is located in the east part of Ulleung Island, which is in the east part of Korea and about 155 km from the mainland. Ulleung Island is 72 km$^2$. The island has a few small mountains whose heights are about 500 to 960 m a.s.l., within 5 km to the north and south-east of the station. There is also a small town in the valley northeast of the station with a small port, which is 810

m away from ULD. Since ULD is on the west slope of a hill, it is mainly affected by airflow up over the hill from the south-west, not from the town in the valley. In the south-west area, there is a small brickyard 200 m from the station. Farming and fishing industries are very active on the island, though there is no farm in the southern area of the island near ULD.

### 2.2     Measurement system: Inlet, drying system and instrument

The measurement system consists of three main parts: inlet, drying system and instruments (Fig. 2). The intake is an inverted stainless steel box (15 cm (W) x 25 cm (D) x 30 cm (H)) with a stainless steel filter (D 4.7 cm, pore size 5 µm) mounted on a plastic mesh holder and connected to the Dekabon sampling tubing (Nitta Moore 1300-10, I.D 6.8 mm, O.D 10 mm). Over times longer than one month, a significant pressure drop occurs between the inlet and the pump, so the filter is replaced monthly.

Sample air is dried with a system that has a cold trap (CT-90, Operon, Korea), which is connected to the pump (KNF

N145.1.2AN.18, Germany, 55 L/min, 7 bar in AMY; KNF N035AN.18, Germany, 30 L/min, 4 bar in JGS and ULD). The cold trap can cool to – 90 °C, however, it is set to – 80 °C so the real temperature of the inner air flow is – 50 °C according to our experiment. When the cold trap is turned on, the air move through a first chamber which drops it to – 20 °C, and then to a second chamber that cools it to – 50 °C. To increase dehumidification efficiency, the second chamber is filled with stainless steel beads (Fig. 2).

Each trap is employed drying ambient air for 24 hr while the other is warmed and drained. The dehumidification and water drain sequence is as follows: (step 1) pump/cold trap A is employed to dry ambient air for 24 hours (step 2) pump/cold trap B turn off to melt ice at ambient temperature for 20 hours (step 3) pump B turns on to pressurize and water drain for 2 hours (step 4) cold trap B turns on and cools to operating temperature for 2 hours (step 1) pump/cold trap B are employed drying ambient air. The difference between this system and a typical cryogenic one is that it was designed with a dual mode, with one trap drying while

water is automatically drained from the other. Therefore it avoids the cold trap impinger clogging during long-term, unattended monitoring. This drying system was developed by KMA and Korea Research Institute of Standards and Science (KRISS) in 2011 for the remote monitoring stations so that it can be easily accessed remotely.

Even though the H$_2$O monitored by CRDS was not calibrated, hourly mean H$_2$O through the drying system is 0.004 ± 0.005%

at AMY, 0.001 ± 0.002% in JGS and 0.001 ± 0.004% in ULD during 2012 to 2016. Laboratory standard gases prepared by

the Central Calibration Laboratory (CCL), which is operated by the National Oceanic and Atmospheric Administration, Global Monitoring Division in Boulder, Colorado, USA, typically contain less than 0.0001% H$_2$O




(www.esrl.noaa.gov/gmd/ccl/airstandard.html). When we sampled them directly to CRDS without this drying system, mean $H_2O$ (10 min average) was 0.0009% regardless of the $CO_2$ level across the KMA monitoring stations.

For example, when there is a difference in $H_2O$ at AMY between laboratory standard gases and ambient samples of 0.003%, this creates a small bias of 0.012 ppm on 400 ppm $CO_2$ according to the equation suggested by Rella et al. (2013):

$$\frac{C_{dilution}}{C_{dry}} = 1 - 0.01 H_{act} \tag{1}$$

where $C$ is the $CO_2$ mole fraction and $H_{act}$ is the actual water mole fraction (in %). Since working standards showed almost same level of $H_2O$ to laboratory standards through the CRDS, we considered the $CO_2$ mole fraction dilution offsets between calibration standards and sample air when the uncertainty was estimated (sect 3.1).

After the drying system, ambient air flows through the 1/8" (o.d.) stainless steel tubing to an 8 port multi-position valve (VICI), which selects among standard gases and ambient air. A leak test of all lines is performed every month. CRDS is well-known for its highly linear and stable response (Crosson, 2008). At AMY, a non-dispersive infrared analyzer (Ultramat 6, Siemens, Germany) was used to monitor atmospheric $CO_2$ every 30 sec from February 1, 1999 to December 31, 2011. A model G2301 (Picarro, USA) was installed in Oct, 2011, and it became our official $CO_2$ measurement at AMY starting January 1, 2012. Picarro

models G1301 and G2401 have been used to measure ambient $CO_2$ and $CH_4$ since January 1 and February 12 in 2012, at JGS and ULD, respectively. Those analyzers monitor $CO_2$ every 5 sec across the KMA network.

### 2.3     Calibration, quality control and data processing

### 2.3.1 Calibration method

Analyzers are calibrated every two weeks with 4 standard gases from 360 to 460 ppm at intervals of 30 ~ 40 ppm. Since AMY is a central lab for the KMA Greenhouse Gas (GHG) network, working standards used at JGS and ULD are filled at AMY and calibrated for $CO_2$ dry air mole fractions traceable to the WMO-X2007 scale. AMY directly calibrates its instrument with laboratory standards certified by the CCL. Laboratory standards have an uncertainty of ± 0.02 ppm and the uncertainty of

working standard is ± 0.05 ppm after transferring the scale. This value is also used as the scale propagation factor of the

measurement uncertainty in section 3.1. When the scale is propagated and an instrument is calibrated, all 4 cylinders are sampled by CRDS for 40 min. The first 30 min of each cylinder run are rejected and 10 min are used for the calibration of $CO_2$ to ensure instrument stabilization. Calibration connects analyzer response to the WMO-X2007 scale, and also tracks drift in the analyzer. The drift of the CRDS over two weeks is normally within ± 0.1 ppm for all KMA stations while 4 standards are adequate to determine $CO_2$, as indicated by mean residuals of 0.0003 ± 0.026 ppm from a linear function fitted to the measurements of

standards. Our ability to maintain and propagate the WMO-X2007 scale was shown through the 6th Round Robin comparison of standards hosted by the CCL (https://www.esrl.noaa.gov/gmd/ccgg/wmorr/wmorr_results.php), a comparison of continuous measurements with the traveling instrument of the World Calibration Centre (WCC-Empa, 2017(a), (b) and 2014.), and a co-located comparison of discrete samples collected at AMY and analyzed by NOAA/ESRL (Fig. 3) with our in situ analyzer results. This ongoing comparison between level 1 (L1) hourly data from the CRDS and weekly flask-air samples collected at AMY has





been implemented since December, 2013. The mean difference between flask minus in situ is -0.11 ± 2.32 ppm from 2014 to 2016, close to GAW's compatibility goal for $CO_2$ in the Northern Hemisphere (± 0.1 ppm) (Fig 3).

**2.3.2 Data quality control process**

All data are monitored, collected and stored at the Environmental Meteorological Research Division (EMRD), National Institute of Meteorological Sciences (NIMS) in Jeju, Korea. Raw data based on 5 second intervals are processed two ways: 1) auto flagging and 2) manual flagging. Auto flagging identifies instrument malfunction and instrument detection limit of $CO_2$. Auto flags are assigned when our algorithm detects deviations from prescribed ranges for analyser engineering data.

Acceptable values for the parameters related to instrument function are: $H_2O$ (%) < 0.02; 139.95 < cavity pressure (Torr) < 140.05; and 44.99 < cavity temperature (℃) < 45.01. $H_2O$ > 0.02% indicates periods when the drying system had problems or a leak in the gas line occurred, while the ranges of cavity pressure and temperature were suggested by the manufacturer. Instrument measurement range is based on the calibration range, from 360 to 460 ppm at 30 - 40 ppm intervals. Therefore flags are assigned when $CO_2$ is outside this range. In the case when $CO_2$ is greater than 500 ppm, values are considered invalid, since a leak in a sample line or pump diaphragm causes measurement of room air.

Manual flags are assigned by technicians at each station according to the logbook based on: inlet filter exchange, diaphragm pump error, low flow rate, dehumidification system error, calibration periods, experimental periods such as participation in comparison experiments, observatory environmental issue such as construction next to a station, extreme weather, or other issues related to the instrument. These codes refer to definitions by the World Data Centre for reactive gases and aerosols maintained by EBAS for the GAW Programme (http://www.nilu.no/projects/ccc/flags/flags.html) and were modified for the Korea network. Data with flags are reviewed by scientists at the EMRD, and valid data are selected as Level 1 (L1).

**2.3.3 Regional background selection method**

L1 data include local pollution by human and/or biotic activities. Therefore, only those data that represent non-polluted and well-mixed air should be selected for analysis on a regional scale. The data are selected as background when they meet the following conditions: 1) Hourly averages are calculated when there are at least 60 30 sec measurements from the NDIR and at least 300 5 sec measurements from the CRDS, 2) the hourly average of level 1 has a standard deviation less than "A", 3) and the differences between consecutive hourly averages are less than "B". A and B were determined empirically and are equal. We determined 1.8 ppm for AMY, 1 ppm for JGS, and 0.8 ppm for ULD. This process selects 55% to 60% of the data at each station, and they are defined as Level 2 (L2) hourly data. To calculate daily averages (L2 daily), at least 6 L2 hourly data are required. Fig. 3 shows L1 hourly data, L2 daily data, and the smoothed curves fitted to L2 daily data calculated with methods by Thoning et al. (1989).

**3.    Results and Discussion**

**3.1    Measurement uncertainty**

Variability in $CO_2$ observed at KMA's stations includes contributions from natural atmospheric variability and variability related to the air handling and measurement procedures. Natural atmospheric variability is represented, for example, by the standard deviation of all measurements contributing to a time-average, after the contribution of experimental noise is accounted for. Here





we develop methods to calculate practical realistic measurement uncertainties. Based on measurements of target cylinders and a co-located comparison of measurements at AMY, we assume systematic biases are negligible. According to the previous studies, the total measurement uncertainty consists of multiple uncertainty components (Andrews et al., 2014, Verhulst et al., 2017). However, in this paper, we assess the measurement uncertainty based on the following components:

$$(U_T)^2 = (U_{h2o})^2 + (U_P)^2 + (U_r)^2 + (U_{scale})^2 \qquad (2)$$

where $U_T$ is the total measurement uncertainty in the reported dry-air mole fractions; $U_{h2o}$ is the uncertainty from the drying system; $U_p$ is repeatability; $U_r$ is reproducibility; and $U_{scale}$ the uncertainty of propagating the WMO-XCO$_2$ scale to working

standard gases.

$U_{h2o}$ is computed from the differences in H$_2$O (%) between the ambient airstream through the drying system and standard gases injected directly, bypassing the drying system. According to the GAW recommendation, the standard gases should be treated through the same system to air sample (WMO, 2016). However, our drying efficiency is not constant so that we injected standard gases directly as a reference value. Here, we define H$_2$O from the standard gases as 0.0009%. This value has been constant and

stable during 2012 to 2016. On the other hand, the drying system efficiency is not constant so this uncertainty component is time dependent. Eq.(1) was applied for this factor where $H_{act}$ is the difference between H$_2$O in samples and standard gases (0.0009%). Hourly CO$_2$ dilution offsets range from -0.05 to 0.09 ppm at AMY, -0.02 to 0.07 ppm at JGS and -0.05 to 0.08 ppm at ULD during 2012 to 2016. Since positive and negative values are found, we use following equation:

$$U_x = \sqrt{\frac{\sum_{i=1}^{N}(x_i)^2}{N}} \qquad (3)$$

where $U_x$ represents $U_{h2o}$; x is the hourly CO$_2$ dilution offsets from Eq(1); N is the total number of hourly mean values. $U_{h2o}$ is tabulated for each station in table 2.

$U_p$ is determined from the standard deviations of working standard measurements, as described in section 2.3.1 and expressed by

a pooled standard deviation

$$U_p = \sqrt{\frac{\sum_{i=1}^{N} N_i \times S_i^2}{N_i - N_t}} \qquad (4)$$

where $S_i$ is the standard deviation of 10 min averages of working standard measurements; $N_i$ the number of data during 10

minutes (based 5 sec intervals); and $N_t$ is the total number of calibrations during the period. $S_i$ varied from 0.02 to 0.09 ppm at AMY, 0.02 to 0.07 ppm at JGS and from 0.01 to 0.05 at ULD. The pooled standard deviations ($U_p$) are shown in table 2.

$U_r$ is the drift occurring between two-weekly calibration episodes, which was mentioned in section 2.3.1. We determined it as the differences in CO$_2$ measured from cylinders with subsequent calibrations over two weeks. It ranged from -0.08 to 0.1 ppm at AMY, -0.07 to 0.09 ppm at JGS and -0.16 to 0.11 ppm at ULD. We expressed $U_r$ as the standard deviation of all drift values

during the experimental period using Eq (3), where $U_x$ represents $U_r$; x $\Delta$CO$_2$ during 2 weeks; and N is the total number of data. They are tabulated with other uncertainty terms by site in table 2.



According to the Zhao et al.,(2006) the uncertainty of working standards can be calculated by the propagation error arising from the uncertainty of primaries with maximum propagation coefficient ($\gamma$ = 1) and repeatability. Similarity $U_{scale}$ for working standard is determined by

$$U_{scale} = \sqrt{U_p^2 + U_{lab}^2} \qquad\qquad (4)$$

where $U_{lab}$ is the uncertainty of laboratory standards, which CCL (NOAA/ESRL) certified. Here, $U_{lab}$ is same value to the uncertainty of Secondaries, 0.070 ppm, in the one-sigma absolute scale. These values are the same for all stations since they are calibrated by a central lab in AMY. Therefore $U_p$ is the repeatability at AMY since we propagate the standard scale through the same anlayzer and set-up for the atmospheric monitoring.

In the future those uncertainties can be greater than now as including more error sources. And also repeatability and reproducibility can be more precise as reflecting variation depending on time.

### 3.2 CO$_2$ data from 2012 to 2016 at KMA's three monitoring stations

The L1 hourly data, L2 daily data and smoothed curves fitted to L2 daily data are shown in Fig. 3. Episodes of elevated CO$_2$ were often observed at AMY, with a mean difference between maximum and minimum L1 hourly values in a year of ~102.1 ± 12.1 ppm; for the other sites, maximum minus minimum values were ~62.5 ± 9.2 ppm at JGS, and ~55.1 ± 9.6 ppm in ULD. The enhancement relative to the local background mole fraction helps evaluate local additions of CO$_2$, with the excess signal defined as:

$$CO_{2XS} = CO_{2OBS} - CO_{2BG}$$

Where $CO_{2OBS}$ is L1 hourly data and $CO_{2BG}$ indicates local background at the site, determined from the smoothed curve fitted to L2 daily data (section 2.3.3). Monthly mean $CO_{2XS}$ at AMY was 4.3 ± 3.3 ppm, with 1.7 ± 1.3 ppm at JGS and 1 ± 1.9 ppm at ULD during 2012 to 2016. As described in section 2.1, since there are a lot of local activities around AMY, the mean value is larger than at other stations. It was assumed that $CO_{2XS}$ is greater in winter compared to other seasons since photosynthesis is not active and respiration is diminished while anthropogenic sources such as residential sectors would dominate. However, all three stations showed highest $CO_{2XS}$ in summer (JJA); it was 6.3 ± 4.9 ppm at AMY, 2.8 ± 1.4 ppm at JGS and 1.6 ± 2.7 ppm at ULD. Meanwhile the smallest $CO_{2XS}$ was during spring (MAM) at AMY with 2.8 ± 1.5 ppm, and during winter (DJF) at JGS and ULD with 0.9 ± 0.5 ppm and 0.4 ± 0.4 ppm respectively. Even though the baseline data, which agree with the conditions given in 2.3.3, accounted for 55% to 60% of total data, the percentages are different according to the seasons. For example, during summer they decreased to 46% at AMY, 43% at JGS and 34% at ULD, meanwhile they account for 61% - 75% at all stations during winter. Since Korea Peninsula is affected by Siberian high from winter to spring with strong westerly wind, $CO_{2OBS}$ was measured in well mixed air relative to summer. Also, the diurnal variation increased during summer, so this statistical selection method is limited. We also discuss this issue in sections 3.3 and 3.4.



### 3.3 Local/regional effects on observed $CO_2$

To understand the influence of local surface wind on observed $CO_2$, bivariate polar plots were used. These plots are expressed by dependence of all hourly $CO_2$ mole fractions on wind direction and speed in 2016 (Fig. 4 to 6). We divided data into two groups: a) selected hourly baseline data (L2 hourly data) from L1 hourly data with conditions suggested in section 2.3.3 (Fig 4-6, left

panel) and b) remaining data after baseline selection (L1 minus L2 hourly data) (Fig 4-6, right panel). An automatic weather station (AWS) was installed at AMY near the inlet, and 10 m above the station at JGS and ULD, but separate from the air inlet tower.

At AMY, baseline $CO_2$ from autumn to winter occurred when winds mainly come from 315° to 360°. In spring, baseline $CO_2$ started to include winds from 180° to 225° and the dominant baseline wind direction shifted to the south (180° to 225°) in

summer, indicating that baseline $CO_2$ is linked to air masses from the sea (Yellow Sea). However, when wind speed is less than 5 $m \cdot s^{-1}$, $CO_2$ is elevated in all seasons.

JGS observed the strongest winds among the three stations for all seasons, with wind speed >7 $m \cdot s^{-1}$ occurring almost 36% of the time and a maximum speed up to ~40 $m \cdot s^{-1}$. Baseline $CO_2$ was observed with winds from 315° to 340° (Yellow Sea) and 120° to 160° (East China Sea) with wind speed > 5 $m \cdot s^{-1}$. In contrast, JGS is contaminated with local $CO_2$ emissions when wind comes

from 45° to 135° with wind speed ≤ 5 $m \cdot s^{-1}$. Since National Geo Park is east of the station, JGS could be affected by tourist activities. During the period from autumn to winter, high $CO_2$ was also often observed with strong wind.

For ULD, the main wind directions are quite clearly from 0° to 90° (30%) and from 180° to 270° (33%), and wind speed less than 5 $m \cdot s^{-1}$ occurs 72% of the time. Since ULD is located on the west slope of a hill, it is only affected by downslope winds from the top of the hill with a northeast direction and upslope winds with a southwest direction. Normally baseline $CO_2$ is

monitored regardless of wind direction and wind speed. High $CO_2$ episodes were mainly observed when the wind sector was between 180° to 225°, presumably affected by the brickyard, 200 m from the station.

Overall, both stations on the west side of Korea, AMY and JGS, are downwind from East Asia so their observations contain information about its sources and sinks when observing baseline $CO_2$, while they are also affected by local activities under stagnant conditions. Our eastern station, ULD, reflects baseline $CO_2$ more than other two stations with limited local activities.

And it was also suggested that data from regional GAW stations have complex information, so it is necessary to develop a selection method for baseline conditions to better understand regional characteristics.

### 3.4 Average diurnal variation

Diurnal $CO_2$ variations, calculated as the average departure from the daily mean, in April, August, November and January, are used to represent the average diurnal variations in spring, summer, autumn and winter over 5 years in Fig 7. The standard

deviations of the hourly means are ~ 16 ppm, ~ 7 ppm and ~ 5 ppm in AMY, JGS and ULD in January, April and November, but increased in August to ~ 20 ppm, ~10 ppm and ~ 8 ppm at AMY, JGS and ULD, respectively.

Many factors, such as the intensity of photosynthesis, density of vegetation and the degree and speed of atmospheric mixing, influence the diurnal fluctuations in atmospheric $CO_2$ (Pales and Keeling, 1965; Inoue and Matsueda, 1996). Generally, rapid growth of turbulence at the surface after sunrise results in a high boundary layer and leads to decreased $CO_2$ measured at the

station during daytime, while $CO_2$ accumulates in a stable nocturnal boundary layer created by a temperature inversion due to surface radiative cooling during the night (Higuchi et al., 2003). Also, the diurnal cycle in summer is the result of a combination of several factors, including active photosynthesis.



AMY and JGS showed those typical characteristics during all seasons, even though the differences between minimum and maximum $CO_2$ values significantly varied with month. However, ULD had this trend only in summer while other seasons showed very steady values through the day.

At AMY, the differences between maximum and minimum values were 13.5 ppm and 6.9 ppm in August and November, respectively, while those values were around 3 ppm in other seasons. This trend is very typical, as mentioned above. For JGS, those values were observed in the order of 9.6 ppm > 3.3 ppm > 2.8 ppm > 0.88 ppm in August, April, November and January, respectively. During summer, both AMY and JGS show an afternoon plateau in $CO_2$ from around mid-afternoon due to the combination of changes in the photosynthetic rate and increased boundary layer before sunset. In the evening $CO_2$ increases again when respiration dominates and the boundary layer becomes neutral or stable.

In contrast, at ULD, an average diurnal cycle was only obvious in August (peak to peak value of 3.9 ppm), meanwhile, $CO_2$ increased without a period of constant $CO_2$ during the afternoon seen at the other two stations. At night time, ULD has downslope winds because the radiative cooling of the hill leads air to flow down the slopes of the station while daytime warming brings the air from the lower slopes of the hill. ULD, at 221 m, is higher than AMY and JGS, so that it is hardly affected by vegetation, but it depends on wind direction according to the mountain and valley breezes. In other seasons, diurnal variations were 0.5~1 ppm.

### 3.5 Seasonal cycle and growth rates in East Asia

Seasonal variations from KMA's three stations and two other stations, WLG and RYO in East Asia, are compared in Fig. 8. WLG flask-air data from NOAA/ESRL/GMD and quasi-continuous measurements at RYO by Japan Meteorological Agency, which were downloaded from the World Data Centre for Greenhouse Gases (WDCGG), were fitted with smoothed curves and compared to KMA observations. It is known that the seasonal cycle of atmospheric $CO_2$ at surface observation stations in the Northern Hemisphere is driven primarily by net ecosystem production fluxes from terrestrial ecosystems (Tucker et al., 1986, Fung et al., 1987, Keeling et al., 1989). The averaged seasonal amplitude from 2012 to 2016 was smallest at WLG with 12.2 ± 0.9 ppm and largest at AMY with 15.4 ± 3.3 ppm. For JGS and RYO, peak to peak amplitudes were similar at 13.2 ± 1.7 ppm and 13.5 ± 1.6 ppm, whereas it was 14.2 ± 3.1 ppm at ULD (Table 3).

Normally, maximum $CO_2$ appears between 4.8 to 5.8 ppm in April while the minimum appears in August between -6.8 to -9.6 ppm according to the station. The highest maximum and lowest minimum mean value appeared at AMY indicating that even though AMY is located at similar latitude as these other stations, it seems to capture photosynthetic uptake and respiration release of $CO_2$ by terrestrial ecosystems more than others. Also atmospheric $CO_2$ at AMY includes added anthropogenic emissions transported from East Asia as explained in section 3.3. Meanwhile WLG is hardly affected by vegetation due to its altitude (Table 1).

The annual growth rate of $CO_2$, which was computed by the increase in annual means of de-seasonal trends from one year to the next at KMA sites, was quite similar to other East Asian stations and to the global growth rate from WMO (Fig.8 (b)). From 2012 to 2016, the average annual increase observed at all stations in East Asia was between 2.4 ± 0.7 and 2.6 ± 0.9 ppm/yr.

This mean value is greater than 1.7 ppm from 1988 to 1998 reported by RYO and is similar to the global increase of 2.21 ppm/yr from 2007 to 2016 reported by WMO (This value is determined by the absolute differences from previous year). The large increase in 2016 and 2015 was due to increased natural emissions of $CO_2$ related to the most recent El Niño event (Betts et al.,




2016). Averaged annual $CO_2$ was highest at AMY and lowest at WLG among East Asian stations listed in Table 3, which shows that their differences are 8.5 ± 0.7 ppm.

Since the residence time of $CO_2$ is long enough in the atmosphere, the growth rate should be similar between the stations even though their levels are different according to those scales and locations. However, our long term trends comparison showed that
measurement and environmental changes also effected on its growth rate.

The long-term trends of $CO_2$ mole fractions at AMY, WLG and RYO from 2002 to 2016, which were extracted by the method of Thoning et al. (1989), are shown in Figure 9. The trends of $CO_2$ at WLG and RYO increased in parallel, whereas AMY increased with a similar slope but with larger fluctuations than the other stations. Especially the negative growth rate, which was only observed in northern high latitude in 1992 due to Mount Pinatubo eruption, was recorded in 2004 and 2006 at AMY, while high
growth rate was recorded in 2012 without ENSO (WDCGG, 2017; Stenchikov et al., 2002; Heimann and Reichetein, 2008).

In July 2004, the inlet height at AMY was changed from 20 m to 40 m above ground; observed $CO_2$ mole fractions before moving the inlet height reflected more influence from local activities that affected the long-term trend (Song et al., 2005). According to the log book, in 2005 AMY was under the construction to expand the space with a new building that the instrument showed unstable signals during the period.

The measurement system was changed in 2012. An NDIR was used to monitor atmospheric $CO_2$ from 1999 with a three step dehumidification system, 1) - 4℃ cold trap 2) nafion and 3) $Mg(ClO_4)_2$, before installing the new system in Oct. 2011 as described in section 2.2. It was proved that the CRDS has higher precision measurements than NDIR, and there were $CO_2$ offsets in a comparison between the two instruments (Chen et al., 2010; Zellweger et al., 2016).

Standard scale was also changed at the same time; KRISS scale was used during the NDIR period and then changed to WMO
X2007 scale with the new system. KRISS and WMO scales agreed well in CCQM-P41 organized by the International Bureau of Weights and Measures (BIPM) (www.bipm.org/utils/common/pdf/final_report/QM/P41/CCQM-P41_part1.pdf). However, maintaining traceability to the primary standard of the same scale under the GAW Programme would be more incentive to assure the long-term consistency (WMO, 2017).

This result suggests that factors not only related to local sources/sinks, but also environmental changes around stations and level
of technical skill are very important to monitor regional background $CO_2$ over the long term. On the other hand, on-going comparisons of measurements were emphasized at a co-located site and for the same species, such as between discrete samples and continuous measurement (Masarie et al., 2001). After 2012, long-term trends increased in parallel, with AMY 5.5 ± 0.3 ppm greater than RYO, and RYO 2.9 ± 0.3 ppm greater than WLG.

## 4. Summary and Conclusions

Now many scientists are on the way to determine regional/national emissions through top-down methods with in situ data, so the importance of high density monitoring stations such as WMO/GAW regional stations is increasing since their data include a lot of information about $CO_2$ fluxes. In this regard, it remains a challenge for WMO/GAW stations to provide high quality data to better constrain emissions and sinks. In this paper we introduced the three KMA stations and measurement systems for high quality data, and we analysed observed $CO_2$ characteristics with comparisons to other East Asia stations.

KMA instrumented three monitoring stations covering the Korean Peninsula in 2012 with a CRDS and a new drying system at each station. The drying system showed 0.001 to 0.004% water vapour in CRDS when sampling of ambient air, while it was



0.0009% in laboratory cylinders; those values satisfy GAW recommendation, 0.0039% (WMO, 2016). It also suggests the possibility to monitor atmospheric species in humid areas with easy maintenance and remote control of the system.

From 2012 to 2016, our measurement uncertainties, which include components of the drying system, measurement repeatability, reproducibility and scale propagation, are quite similar with 0.116 ppm, 0.114 ppm and 0.114 ppm at AMY, JGS and ULD

respectively. In the future those uncertainties may increase as other components of uncertainty, and their variations over time, are added.

We assessed the $CO_2$ enhancement using local background level at each station; those were 4.3 ± 3.3 ppm at AMY while 1.7 ± 1.3 ppm at JGS and 1 ± 1.9 ppm at ULD during 2012 to 2016. This indicates that AMY has high $CO_2$ episodes compared to the other stations. Selected $CO_2$ mole fractions observed at AMY and at JGS in the west part of Korea are more sensitive to East

Asia (e.g., China) according to wind direction and speed. Meanwhile they also reflect regionally contaminated $CO_2$ under the stagnant conditions. At JGS, however, local anthropogenic emissions were very limited and the long-transported $CO_2$ levels are lower compared to AMY due to its high wind speed for all seasons. The diurnal variations at these two stations indicate they reflect the impacts of local vegetation and the degree and speed of atmospheric mixing. ULD, east of the Korean mainland, observed well-mixed air masses with small diurnal variations in $CO_2$ and similar $CO_2$ levels regardless of wind direction and

speed. Due to its locations it is affected by mountain and valley breezes mainly.

The seasonal variation at AMY is large compared to the other stations in East Asia, indicating that it could be affected by not only vegetation but also added anthropogenic emissions transported from East Asia. $CO_2$ observed at three KMA stations is higher than at WLG and similar to RYO as expected by their locations, while for growth rate, they are very similar to RYO and WLG during 2012 to 2016.

When AMY was compared to WLG and RYO in East Asia over 15 years, the long-term trend increased with a similar slope but with larger fluctuations compared to the other two stations. This seems to reflect not only carbon sources and sinks but also environment changes at the stations and level of sophisticated measurement expertise.

Since $CO_2$ observed in KMA includes much information about carbon fluxes in East Asia, these data are helpful to improve understanding of the carbon cycle in this region. In addition, to enhance the understanding of $CO_2$ observations at Korean

monitoring stations, isotopes measurements such as $^{14}C$ in $CO_2$ would be very useful (Turnbull et al., 2011).

*Acknowledgments.* We would like to thank Dr. Edward Dlugokencky (NOAA/ESRL) for his helpful comments on this paper. We also thank Ryori station in Japan and Waliguan station in China for their data contributions. Finally we appreciate all staff and technicians at AMY, JGS, and ULD in Korea network. This work was funded by the Korea Meteorological Administration
Research and Development Program "Research and Development for KMA Weather, Climate, and Earth system Services– Support to Use of Meteorological Information and Value Creation" under Grant (KMA2018-00521).





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





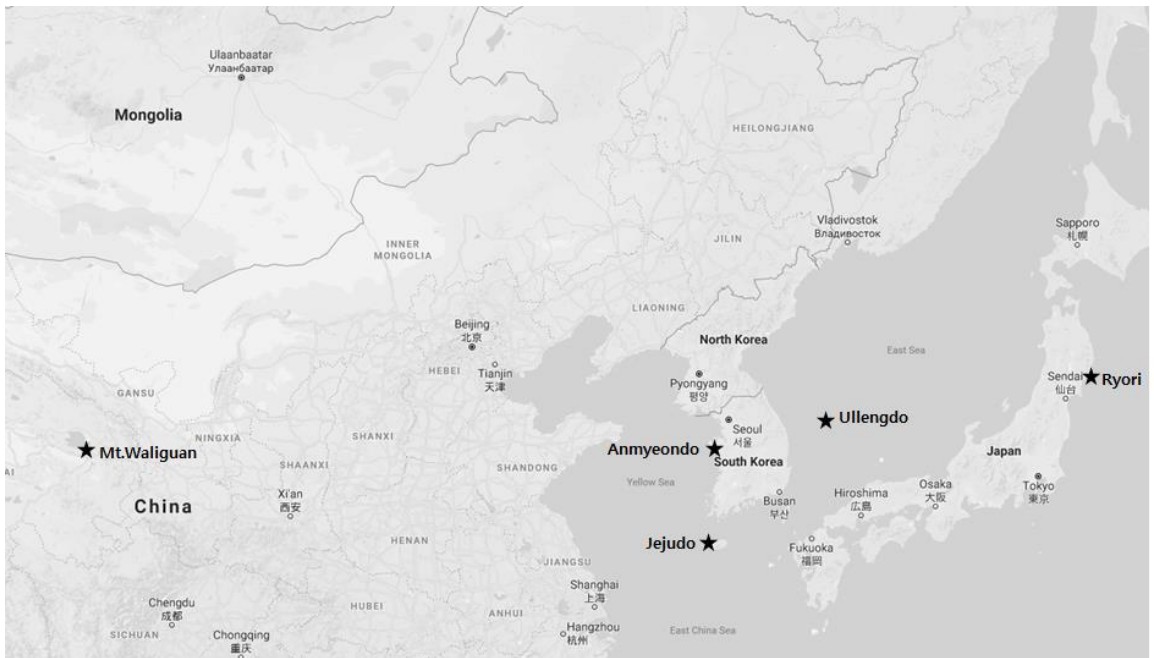

Figure 1. Locations of the three KMA monitoring stations in Korea, and Mt.Waliguan WMO/GAW global station and Ryori WMO/GAW regional station in East Asia.




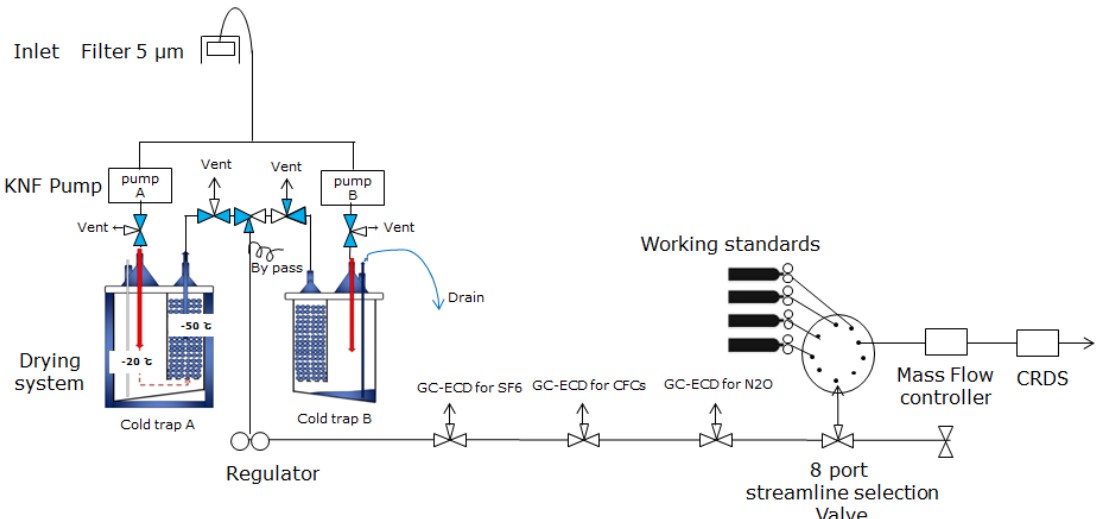

Figure 2. Schematic of the in-situ system when the drying system is at the state of step 3 in AMY, JGS and ULD.




(a)

(b)

(c)

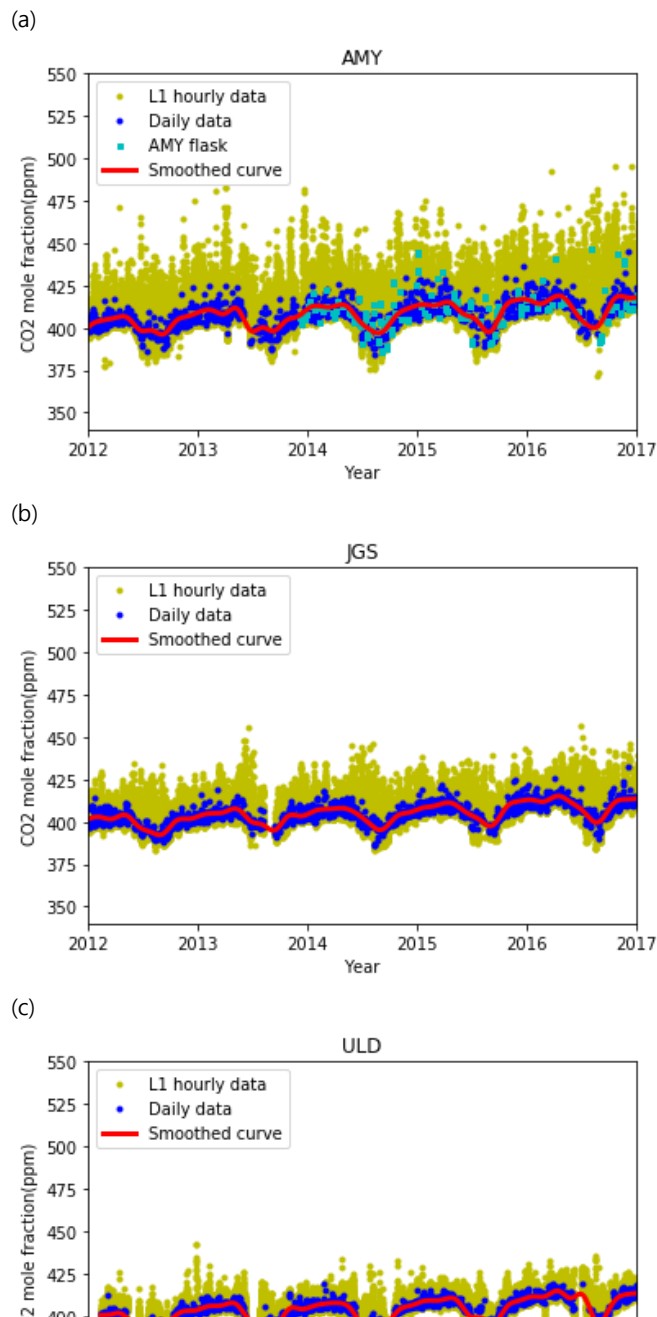

Figure 3. L1 hourly (yellow dots, $CO_2$ OBS), L2 daily (blue dots) averaged, and smoothed curves fitted to L2 daily averages (red line, $CO_2$ BG) at (a) AMY, (b) JGS and (c) ULD.





Figure 4. Bivariate polar plots for (left panel) selected baseline $CO_2$ (L2) and (right panel) polluted $CO_2$ in spring (L1-L2) (a and b), summer (c and d), autumn (e and f), and in winter(g and h) at AMY in 2016.





Figure 5. Bivariate polar plots for (left panel) selected baseline $CO_2$ (L2) and (right panel) polluted $CO_2$ in spring (L1-L2) (a and b), summer (c and d), autumn (e and f), and in winter (g and h) at JGS in 2016.



Figure 6. Bivariate polar plots for (left panel) selected baseline $CO_2$ (L2) and (right panel) polluted $CO_2$ in spring (L1-L2) (a and b), summer (c and d), autumn(e and f), and in winter(g and h) at ULD in 2016.



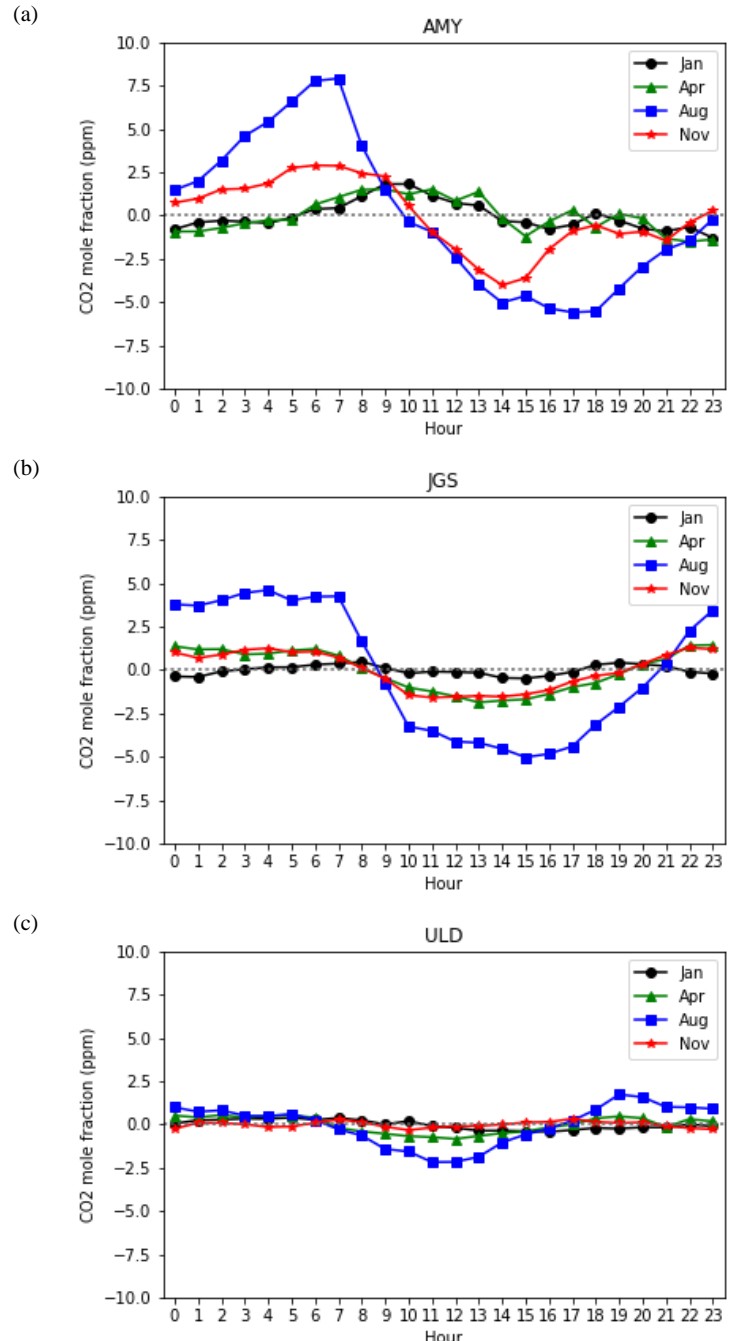

Figure 7. Mean diurnal variations of $CO_2$ mole fraction. Values show the average departure from the daily mean in January, April, August and November at (a) AMY, (b) JGS and (c) ULD from 2012 to 2016.





(a)                                                              (b)

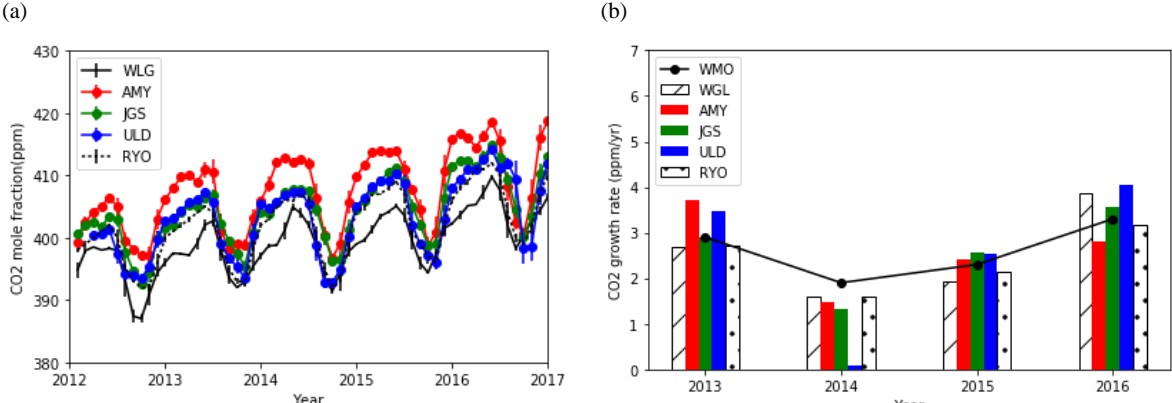

Figure 8. The time series of (a) the monthly mean $CO_2$ and (b) the annual growth rate at WLG, AMY, JGS, ULD and RYO. Annual growth rate was defined as the increase in the annual mean of de-seasonal (long term trend) values from the corresponding value in the previous year. The growth rate reported by WMO is overlaid on (b) and this value is annual increase (not de-seasonal), absolute differences from the previous year.



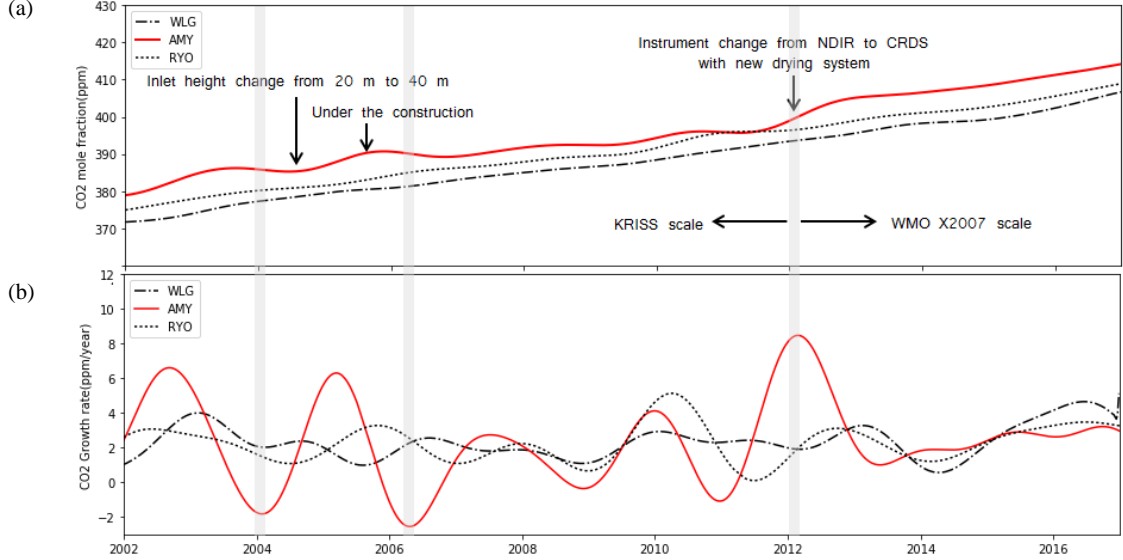

Figure 9. (a) Long-term trend of atmospheric $CO_2$ and its (b) instantaneous growth rate at WLG, AMY and RYO. Overlaid grey line indicated the period of the negative (in 2004 and 2006) and positive (in 2012) growth rates at AMY compared to other two East Asia stations (WLG and RYO).





Table 1. Information about the three KMA monitoring stations in Korea and the two monitoring stations in East Asia

| Station | ID | Longitude | Latitude | Altitude | Inlet height | Measurement History |
|---|---|---|---|---|---|---|
| Anmyeondo, Korea | AMY | 126.32°E | 36.53°N | 47 m | 40 m | Since 1999 |
| Jejudo Gosan Suwolbong, Korea | JGS | 126.16°E | 33.30°N | 71.47 m | 6 m | Since 2012 |
| Ulleungdo*, Korea | ULD | 130.90°E | 37.48°N | 220.9 m | 10 m | Since 2012 |
| Mt.Waliguan, China | WLG | 100.90°E | 36.28°N | 3810 m | 5 m | Since 1990 |
| Ryori, Japan | RYO | 141.82°E | 39.03°N | 260 m | 20 m | Since 1987 |

*ULD is not GAW station.





Table 2. The uncertainty estimates for measurements of $CO_2$ at each station from 2012 to 2016. Units are ppm. All terms are in the 68% confidence interval

| Uncertainty factors | AMY | JGS | ULD |
|---|---|---|---|
| $U_{h2o}$ | 0.023 | 0.009 | 0.018 |
| $U_p$ | 0.053 | 0.046 | 0.025 |
| $U_r$ | 0.048 | 0.056 | 0.065 |
| $U_{sacle}$ | 0.088 | 0.088 | 0.088 |
| $U_T$ | 0.116 | 0.114 | 0.114 |





Table 3. Annual mean abundances from 2012 to 2016, mean seasonal amplitude and growth rates. The uncertainties are standard deviations during each period. Seasonal amplitudes are calculated from the detrended data. $CO_2$ at ULD, 2012 was calculated only from February to December, without January. Units are dry-air mole fractions (ppm)

| Year | WLG | AMY | JGS | ULD | RYO |
|---|---|---|---|---|---|
| 2012 | 394.7 ± 3.9 | 402.8 ± 3.6 | 399.7 ± 3.7 | 398.4 ± 3.6 | 397.6 ± 3.7 |
| 2013 | 397.2 ± 3.1 | 405.4 ± 4.6 | 402.5 ± 3.5 | 401.8 ± 4.4 | 400.1 ± 4.2 |
| 2014 | 398.6 ± 3.8 | 407.8 ± 5.7 | 403.9 ± 4.0 | 401.9 ± 5.5 | 401.7 ± 5.1 |
| 2015 | 401 ± 3.3 | 410.2 ± 5.7 | 407.0 ± 4.5 | 405.0 ± 5.0 | 404.1 ± 4.4 |
| 2016 | 404.9 ± 3.2 | 412.6 ± 6.1 | 410.0 ± 4.6 | 409.3 ± 5.1 | 407.4 ± 4.5 |
| Mean seasonal amplitude over 5 years. | 12.2 ± 0.9 | 15.4 ± 3.3 | 13.2 ± 1.7 | 14.2 ± 3.1 | 13.5 ± 1.6 |
| *Maximum* | *5.4 ± 0.7* | *5.8 ± 0.7* | *4.8 ± 0.4* | *5.4 ± 1.0* | *5.6 ± 0.4* |
| *Minimum* | *-6.8 ± 0.7* | *-9.6 ± 2.6* | *-8.3 ± 1.3* | *-8.8 ± 2.3* | *-7.9 ± 1.3* |
| Mean annual growth rate over 5 years (ppm·yr$^{-1}$) | 2.5 ± 1.1 | 2.5 ± 0.7 | 2.6 ± 0.9 | 2.5 ± 1.7 | 2.4 ± 0.7 |