# Peer review of "The measurement of atmospheric CO2 at KMA/GAW regional stations, the characteristics, and comparisons with other East Asian sites"

_Atmospheric Chemistry and Physics, 2018_

## Referee Comment (RC1) · Anonymous Referee #1 · 29 Oct 2018

General comments

This paper presents the technical details and data from regional CO2 measurement stations within Korea and within the region. It is a welcome paper at a time when there is significant growth in regional carbon cycle monitoring and interest in the value of the data. In particular, because of Korea's location to the eastern seaboard of China, there could be great interest in these data sets going forward.

I commend the authors for presenting the full time series from their stations, indicating features attributable to technical and changes in the local environment. These lessons are valuable for others establishing regional monitoring stations.

[Figure]

I recommend publication after minor revisions as detailed below.

Specific comments

2.3.1 Calibration method

Fortnightly drift of +/-0.1ppm is significant. Have the authors assessment instrumental drift on shorter timescales? Also, does the calibration correction occur as a step-wise change fortnightly, or is it applied as a linear interpolation between successive calibrations?

P5, line 12: While $CO_2$ > 500ppm will be measured in room air, there are likely many instances (especially at AMY, which is most influenced by the local biosphere) when at night time biospheric respiration coupled to a low boundary layer lead to real $CO_2$ values above 500ppm. Although this may currently be difficult to interpret, model skill is improving and it seems a pity to flag all such data as invalid when it likely reflects an important component of the carbon cycle.

P7, line 2: replace 'Similarity' with 'Similarly'

P7, line 7: I suggest 'Here, U_lab has the same value as the . . .'

P7, line 11: I suggest, 'In future, quote uncertainties could be greater due to including more error sources. Repeatability and reproducibility may become more precise with improvements in technologies and methods.'

P9, line 10: I suggest, '. . .3.9ppm) and $CO_2$ increased monotonically during the afternoon.'

P9, line 34: I find mention of the growth rate at RYO during the 1990s a distraction. I suggest deleting it unless it is discussed further.

P10, line 3: I suggest, 'Since $CO_2$ is a long-lived atmospheric species, the growth rates should be similar between stations in the same region, even if they are subject to different combinations of anthropogenic and biogenic fluxes.' I certainly wouldn't use

the phrase 'those scales' as they should all be on the same mixing ration scale.

P10, line 14, I suggest replacing 'unstable' with 'strong, highly localised'

P10, line 25, I suggest deleting 'On the other hand' and having 'On-going comparisons of measurements at co-located sites and for the same species, such as between discrete samples and continuous measurements (Masarie et al, 2001) are a valuable means to maintain data quality and identify sampling issues rapidly.

Figure 8: It is striking that the growth rate for ULD is so low in 2014, yet it is not commented on in the text. Do the authors have any idea what is responsible for that anomalous year at ULD? Discussion of it in the text would be very interesting.
* * *

---

## Referee Comment (RC2) · Anonymous Referee #2 · 31 Oct 2018

**Review of "The measurement of atmospheric CO₂ at KMA/GAW regional stations, the characteristics, and comparisons with other East Asian sites" by Lee et al.**

**Summary of the manuscript**

The authors have made $CO_2$ measurements using NDIR and CRDS instruments at the 3 Korean stations, AMY, JGS and ULD. They described the measurement system, overview of characteristics of $CO_2$ variations observed at these stations. Comparisons with variations at other stations in East Asia are also given.

**General comment**

Atmospheric measurement is a basis of top-down estimation of $CO_2$ emissions and uptakes. Given the increasing importance of East Asia in the global carbon budget, more number of high-quality measurements in the region helps carbon cycle studies. From this point of view, this work, which provides descriptions of hitherto unpublished $CO_2$ measurements and their characterization from Korea, is well acknowledged. This is a good contribution to the community and well within the scope of the ICDC10/GGMT-2017 special issue of ACP. My comments, which might be considered before acceptance of this manuscript, are detailed below.

1. I assume that "characteristics" and "comparisons" in the title are for variations observed at the different stations, and not for technical aspects of measurements. I would re-consider the title (in particular latter part) to make what is addressed in this manuscript clearer.

2. In the manuscript, the term "calibration" is used in different meanings. Sometimes the term is used to link instrument responses of an analyzer to known values of standards i.e. determination of the instrument response curves. In other places, the term is used to measure laboratory in-house cylinders for $CO_2$ mole fraction against standards at higher hierarchy levels i.e. propagation of $CO_2$ scale values from a standard to a standard. I would suggest to give definition of "calibration" at the beginning of section 2.3.1 and to use it exactly as same throughout the manuscript. It would improve readability of the section. In my understanding, "calibration" is in many cases used with the latter meaning in the WMO/GAW community.

3. In this respect, it is important to clearly describe KMA's standard scale system.

According to the text, "laboratory standards" provided from CCL (NOAA/ESRL) are positioned at the highest hierarchy (how many cylinders covering x ppm to y ppm?). Working standards routinely used at the stations are positioned at the 1 lower level and measured directly against "laboratory standards". After this, sample air at the stations is measured against the working standards. These are fundamental information in maintaining the scale at KMA and in propagating the scale from CCL's primary standard cylinders to sample data in Korea. I suggest the authors to re-structure section 2.3.1 and present such basic information systematically in the very first paragraph. In addition, determination of the response curve of the CRDS instrument and long-term surveillance of instrument condition are different topics, which could come after the description of the standards.

4.  It would be a great help for readers if zoom-in maps of the 3 stations, which illustrate the surroundings of the stations as described in section 2.1, could be presented. The geographical scale of Figure 1 is still good in the context of relatively large-scale variations (e.g. Figs 8 and 9), but for smaller-scale phenomena that appear in Figs 4 to 7, information of surrounding geography play larger roles as discussed in the manuscript.

5.  I think that the idea of "background" and "baseline" data (or $CO_2$ mole fraction) may not be consistent throughout the manuscript. In section 2.3.3, the authors define criterion for selecting "background" data. After that, in section 3.2, the authors define $CO_{2BG}$ which is defined the curve fitting by Thoning et al. I do not think these two "backgrounds" are in agreement. The latter is composed of the long-term trend and seasonal cycle, which reflect global, hemispheric to regional variations. In contrast, the former contains synoptic scale variations, for instance elevated $CO_2$ events caused by tracking cyclones which transport signals from continental $CO_2$ emissions (e.g. Tohjima et al. 2010, 2014). The authors split "local" $CO_2$ elevations and "background/baseline" $CO_2$ level only, but discussions on such synoptic variations (intermediate scale) are missing. Such events are however important in the regional context of monitoring emissions from China, to which the authors mention in introduction and conclusion as value of the dataset. I hope to see, even briefly, discussions on synoptic variations, since it would help future data users who address $CO_2$ emissions from China using the data presented in this

study.

**Specific comments**

P3 L21: "can cool" to "can be cooled"

P3 L21: this sentence might be reformulated to read "…-80° C, which makes the real temperature of inner air flow to be -50° C." The -80° C seems to be a set temperature. Where is the temperature sensor placed?

P3 L22: "drops it" and "cools it"—hard to get what "it" means. If I understand correctly, this sentence might be reformulated for instance, "The sample air is cooled to -20° C in the first trap, and then to -50° C in the second trap."

P3 L25: "One of the dual traps is used to dry ambient air for 24 hr while…"

P3 L25: here "hr" is used instead of "hours" used at other places.

P4 L20: See my comment above. Here the calibration means anchoring the analyzers' responses to the $CO_2$ scale guaranteed by the standards.

P4 L22: See my comments above. Here the calibration means to measure working standards against the $CO_2$ scale.

P4 L22: See my comments above. This sentence is very unclear. Should this be read like "working standards used at AMY are those directly provided by CCL?"

P4 L23: Insert "the" berfore "laboratory standards"

P4 L23: Is the "laboratory standards" primary standards that realize the WMO scale i.e. those at highest hierarchy at KMA? If this is the case, the term like "KMA primary standards" might better clarify the standard category at KMA. How large $CO_2$ mole fraction range is covered by the "laboratory standards"? And are these "laboratory standards" same as the "4 standard gases" appearing at the beginning of the paragraph? In summary, questions are how many "KMA primary standards" (that cover XX to YY ppm in $CO_2$ mole fraction) are prepared and how many "working standards" (that cover XXX to YYY ppm) are prepared for each station? Please describe these information systematically. Also, with what kind of instrument are the working standards measured against laboratory standards?

P4 L25: "When the scale is propagated" – this is normally called "calibration". I understand that, firstly, working standards are calibrated against "laboratory standards" to assign them the WMO $CO_2$ scale values, and secondly, by analyzing these gases, the instrument responses are linked to the WMO $CO_2$ scale values.

P4 L26: Here "calibration" means to determine the instrument response curve.

P4 L27: Again "calibration" is used same as above.

P4 L30: Please exactly indicate the degree of agreement between KMA and CCL (+/-0.0X±0.0Y ppm) found from the Round Robin.

P5 L12: "500 ppm". First, I would expect that some $CO_2$-elevated events (caused by "local" or "regional" sources) where $CO_2$ mole fraction exceeds 500 ppm can happen. This data treatment may perhaps lose data with scientific value. Second, since the atmospheric $CO_2$ mole fraction is increasing, I would use a value that well follows the atmospheric trend for instance XX ppm + the long-term trend. The constant value 500 ppm does not mean same as that in past or future years.

P6 L30: "calibrations"

P7 L24: "1.0±1.9 ppm at ULD"

P7 L25: As in my earlier comment, I need to question if the positive values in $CO_{2XS}$ at the stations are simply ascribed to "local activities".

P8 L5: "An automatic weather station…" This sentence might be moved to section 2.1.

P8 L15: What are the "tourist activities" specifically? Local transportation?

P8 L23: As in my earlier comment, the wording like signals of Chinese emissions in "baseline" data may be debatable. And "downwind of East Asia" – note that these two stations are also in East Asia.

P8 L32: "the degree and speed of atmospheric mixing" If the authors means dynamics of the PBL, they might mention to rectifier effect (e.g. Denning et al. 1999; Chan et al. 2008). The wording might be re-considered. Same comment as P11 L13.

P9 L7: Although I am not a non-native in English, I wonder if "plateau" can represents stabilization after decrease (not increase).

P9 L29: same comment as to P8 L23. AMY is also in East Asia.

P9 L10-15: Unfortunately, this paragraph does not try to explain possible causes of the diurnal variation observed in August. The up-valley and down-valley wind feature was already described in section 2.1. Here I hope to see discussions on how such a wind pattern or any possible sources/sinks upwind could affect variations in $CO_2$.

P9 L25: "4.8 to 5.8 ppm" and "-6.8 to -9.6 ppm" I guess these values are deviations from a certain value like an annual average from each station. Please explain.

P10 L15–23: All detailed technical information should be moved to section 2. In this section the authors should focus on how such technical events affected the

measurement data.

P11 L7: "using" to "relative to"

P11 L10: "regionally" to "locally"

P11 L11: What is "the long-transported $CO_2$ levels"? High wind speed does not explain the relatively low $CO_2$ level.

P11 L15: "Due to its location it is…" My understanding is that the latter "it" means "$CO_2$ mole fraction observed at ULD". If this is the case, this sentence is strange. Mountain and valley breezes cannot change $CO_2$ mole fraction directly.

P11 L17: delete "added"

Table 3 caption: "abundances" to "mole fractions".

Table 3: According to the caption, the uncertainties are simple standard deviations calculated from the all data collected during the respective years. It includes signals of the all components: the long-term, seasonal, synoptic-scale and diurnal variation, and results in too big estimates of uncertainties of the annual mean values. Indeed, the numbers tabulated in Table 3 apparently show that there are no significant differences in annual means between every successive years (i.e. no trend is detectable). The authors should calculate uncertainty that better represent an estimate of error of an average.

References:

Chan et al. 2008, doi.org/10.1029/2007JD009443

Denning et al. 1999, doi.org/10.3402/tellusb.v51i2.16277

Tohjima et al. 2010, doi.org/10.5194/acp-10-453-2010

Tohjima et al. 2014, doi.org/10.5194/acp-14-1663-2014

---

## Author Comment (AC1) · 8 Jan 2019

Authors response to both reviews follows. A copy of the reviewer comment is given (with bullet point 'number') followed by a response (blue font).

Response to referee 1

1. General comments
This paper presents the technical details and data from regional $CO_2$ measurement stations within Korea and within the region. It is a welcome paper at a time when there is significant growth in regional carbon cycle monitoring and interest in the value of the data. In particular, because of Korea's location to the eastern seaboard of China, there could be great interest in these data sets going forward. I commend the authors for presenting the full time series from their stations, indicating features attributable to technical and changes in the local environment. These lessons are valuable for others establishing regional monitoring stations. I recommend publication after minor revisions as detailed below

We very appreciate your comments, and commendation on our paper. According to your specific comments, we revised our manuscript.

2. 2.3.1 Calibration method
Fortnightly drift of +/-0.1ppm is significant. Have the authors assessment instrumental drift on shorter timescales? Also, does the calibration correction occur as a stepwise change fortnightly, or is it applied as a linear interpolation between successive calibrations?

Our instrumental drift is very negligible and +/- 0.1 ppm was maximized value of the drift. This result is in common the sentences in section 3.1 "Based on measurements of target cylinders and a co-located comparison of measurements....are negligible" And also because we consider this drift is small, the calibration correction occurs as a stepwise change rather than a linear interpolation between the calibrations. Therefore we suggested mean values in the manuscript. For the clarification, we also added the sentence how to correct the values through the calibration (stepwise change).
We also revised uncertainty value of laboratory standards and working standard gases from 0.02 to 0.07 ppm and from 0.05 to 0.088 ppm respectively according to the Table 2. Since 0.02 ppm and 0.05 ppm were precisions, it needs to make it clear. This section is re-constructed according to referee 2.

3. P5, line 12: While CO2 > 500ppm will be measured in room air, there are likely many instances

(especially at AMY, which is most influenced by the local biosphere) when at night time biospheric respiration coupled to a low boundary layer lead to real CO2 values above 500ppm. Although this may currently be difficult to interpret, model skill is improving and it seems a pity to flag all such data as invalid when it likely reflects an important component of the carbon cycle.

We totally agree with your comments. And we also cannot define all CO2 > 500 ppm are invalid. Since we use the values of 500 ppm for CO2 (due to human issue) and 3000 ppb for CH4 (due to carrier gas from GC) to detect room air due to line leakage, we commented it. So we removed this sentence and also next sentence reflected this issue as well that it is not necessary to comment again.

4. P7, line 2: replace 'Similarity' with 'Similarly'

Corrected

5. P7, line 7: I suggest 'Here, U_lab has the same value as the . . .'

Corrected

6. P7, line 11: I suggest, 'In future, quote uncertainties could be greater due to including more error sources. Repeatability and reproducibility may become more precise with improvements in technologies and methods.'

Corrected

7. P9, line 10: I suggest, '. . .3.9 ppm) and CO2 increased monotonically during the afternoon.'

Corrected

8. P9, line 34: I find mention of the growth rate at RYO during the 1990s a distraction. I suggest deleting it unless it is discussed further.

Corrected

9. P10, line 3: I suggest, 'Since CO2 is a long-lived atmospheric species, the growth rates should be similar between stations in the same region, even if they are subject to different combinations of anthropogenic and biogenic fluxes.' I certainly wouldn't use the phrase 'those scales' as they should all be on the same mixing ratio scale.

We suggested the "scale" as meaning of "either regional station or background station" However, as you mentioned the word of scale is used as normally calibration scale in the community and also seems to overlap to the next word of 'location'. So we also revised according to your suggestion.

10. P10, line 14, I suggest replacing 'unstable' with 'strong, highly localised'

Corrected

11. P10, line 25, I suggest deleting 'On the other hand' and having 'On-going comparisons of measurements at co-located sites and for the same species, such as between discrete samples and continuous measurements (Masarie et al, 2001) are a valuable means to maintain data quality and identify sampling issues rapidly.

Corrected

12. Figure 8: It is striking that the growth rate for ULD is so low in 2014, yet it is not commented on in the text. Do the authors have any idea what is responsible for that anomalous year at ULD? Discussion of it in the text would be very interesting.

Thank you for the comment on this issue. We had discussed this before submitting this paper but could not find out clear reason. We just could find that the monthly mean from July to August is lower than those values in previous years. It seems to lead similar annual mean in 2014 to that value in 2013 and low growth rate in 2014. On the other hand, we cannot distort our data or delete without certain reasons that we just decided to submit the data itself. After taking your comment, we looked at the data again, checked all flags and any other environmental issues, but still unclear. In this time, we addressed this issue directly in the paper for the readers who are interested in it. We added the sentence: "
[revised manuscript text omitted]

---

## Author Comment (AC2) · 8 Jan 2019

Authors response to both reviews follows. A copy of the reviewer comment is given (with bullet point 'number') followed by a response (blue font).

**Response to referee 2**

1. Atmospheric measurement is a basis of top-down estimation of CO2 emissions and uptakes. Given the increasing importance of East Asia in the global carbon budget, more number of high-quality measurements in the region helps carbon cycle studies. From this point of view, this work, which provides descriptions of hitherto unpublished CO2 measurements and their characterization from Korea, is well acknowledged. This is a good contribution to the community and well within the scope of the ICDC10/GGMT-2017 special issue of ACP. My comments, which might be considered before acceptance of this manuscript, are detailed below.

   We thank Referee 2 for the comments on this paper's value. We also appreciate your helpful comments to improve our manuscript.

2. I assume that "characteristics" and "comparisons" in the title are for variations observed at the different stations, and not for technical aspects of measurements. I would re-consider the title (in particular latter part) to make what is addressed in this manuscript clearer.

   As you mentioned our paper discussed about 'measurement of $CO_2$', 'the data characteristics' and 'comparisons with other stations'. So the title also considers those three issues. "The measurement" means process of experimentally obtaining one or more quantity values that can reasonably be attributes to a quantity (JCGM, 2012). WMO/GAW also has an agreement with BIPM that we assumed this title is appropriate in this time. If we directly address 'measurement techniques' in the title, the meaning cannot explain measurement activities such as the calibrations, data processing, and measurement uncertainties.
   We hope referee can agree this title "The measurement of atmospheric CO2 at KMA/GAW regional stations, the characteristics, and comparisons with other East Asian sites".
   For above-mentioned reason, we revised the sentence, which contains the 'measurement' in the section 1 to make what is addressed in the manuscript more clear.

3. In the manuscript, the term "calibration" is used in different meanings. Sometimes the term is used to link instrument responses of an analyzer to known values of standards i.e. determination of the instrument response curves. In other places, the term is used to measure laboratory in-house cylinders for CO2 mole fraction against standards at higher hierarchy levels i.e. propagation of CO2 scale values from a standard to a standard. I would suggest to give definition of "calibration" at the beginning of section 2.3.1 and to use it exactly as same throughout the manuscript. It would improve readability of the section. In my understanding, "calibration" is in many cases used with the latter meaning in the WMO/GAW community.

The definition of calibration is "operation that, under specified conditions, in a first step, establishes a relation between the quantity values with measurement uncertainties provide by measurement standards and corresponding indications with associated measurement uncertainties and, in a second step, uses this information to establish a relation for obtaining a measurement result for an indication" (JCGM, 2012). This concept is same to WMO/GAW community and other measurement communities.

The calibration also is involved in scale propagation: traceability chain is the sequence of measurement standards and calibration that is used to relate a measurement result to a reference (JCGM, 2012).

So we improve section 2.3.1 in our manuscript with the definition of calibration as making it more clear for readers. Here, we revised uncertainty value of laboratory standards and working standard gases from 0.02 to 0.07 and from 0.05 to 0.088 ppm respectively according to the Table 2. Since 0.02 ppm and 0.05 ppm were precisions, it needs to make it clear. And few sentences are revised according to referee 1.

4. In this respect, it is important to clearly describe KMA's standard scale system. According to the text, "laboratory standards" provided from CCL (NOAA/ESRL) are positioned at the highest hierarchy (how many cylinders covering x ppm to y ppm?). Working standards routinely used at the stations are positioned at the 1 lower level and measured directly against "laboratory standards". After this, sample air at the stations is measured against the working standards. These are fundamental information in maintaining the scale at KMA and in propagating the scale from CCL's primary standard cylinders to sample data in Korea. I suggest the authors to re-structure section 2.3.1 and present such basic information systematically in the very first paragraph. In addition, determination of the response curve of the CRDS instrument and long-term surveillance of instrument condition are different topics, which could come after the description of the standards.

We reconstructed section 2.3.1 according to your comments as including the number of cylinders, covering ranges and order of paragraphs. Please confirm the revised manuscript.

5. It would be a great help for readers if zoom-in maps of the 3 stations, which illustrate the surroundings of the stations as described in section 2.1, could be presented. The geographical scale of Figure 1 is still good in the context of relatively large-scale variations (e.g. Figs 8 and 9), but for smaller-scale phenomena that appear in Figs 4 to 7, information of surrounding geography play larger roles as discussed in the manuscript.

Thank you for the comment and it would be informative to readers. We added the more figures in Figure 1 which show the surrounding environment of each station.

(a)

[Figure]

(b)

(c)

(d)

Figure 1. Locations of (a) the three KMA monitoring stations in Korea, and Mt.Waliguan WMO/GAW global station and Ryori WMO/GAW regional station in East Asia. Surrounding Environment of the (b) Anmyeondo (AMY), (c) Jejudo Gosan Suwolbong (JGS), and (d) Ullengdo (ULD) station. Those figures are derived from Google map.

6. I think that the idea of "background" and "baseline" data (or CO2 mole fraction) may not be consistent throughout the manuscript. In section 2.3.3, the authors define criterion for selecting "background" data. After that, in section 3.2, the authors define CO2BG which is defined the curve fitting by Thoning et al. I do not think these two "backgrounds" are in agreement. The latter is composed of the long-term trend and seasonal cycle, which reflect global, hemispheric to regional variations. In contrast, the former contains synoptic scale variations, for instance elevated CO2 events caused by tracking cyclones which transport signals from continental CO2 emissions (e.g. Tohjima et al. 2010, 2014). The authors split "local" CO2 elevations and "background/baseline" CO2 level only, but discussions on such synoptic variations (intermediate scale) are missing. Such events are however important in the regional context of monitoring emissions from China, to which the authors mention in introduction and conclusion as value of the dataset. I hope to see, even briefly, discussions on synoptic variations, since it would help future data users who address CO2 emissions from China using the data presented in this study.

Thank you for the comments. We agree with your opinion that we tried to separate local/long range transported pollutions and baseline data. And also we keep the consistent meaning of baseline which is the result of fitting curve to reduce the noise due to synoptic-scale atmospheric variability and measurement gaps. We improved our relevant manuscript in section 2.3.3, 3.2 and 3.3.

We had the footprint of CO2xs, which were presented in KMA (2014) and suggested below (Figure A). We did not present the figures on this paper since now we are improving our baseline selection method to apply for those footprints and preparing for another paper. However, we can provide those published figures as supplements if it is necessary. Therefore we mentioned this result briefly in the manuscript about this result in section 3.2.

We revised the Figures 4 to 6 in section 3.3 since in this section L2 hourly data was defined as baseline $CO_2$. Therefore to avoid making readers confused, we showed bivariate polar plots with only L1 data. We also thought this plots are enough to explain local/regional effects on observed $CO_2$. The observed CO2 had been distinguished into baseline and pollutions in this section. However, we categorized into two groups as lower and high CO2 after the revisions.

(a)

(b)

(c)

[Figure]

Figure A. The example of $CO_{2XS}$ footprints at (a) AMY, (b) JGS and (c) ULD from 2012 to 2013. 4 days backward trajectories at 500 m using HYSPLIT 4 model based on wind fields provided by NCEP GDAS. This figure was shown the Korea GAW report published in 2014. This method was applied by Li et al. (2014).

[Figure]

Figure 4. Bivariate polar plots for observed CO$_2$ (L1) in (a) winter, (b) spring, (c) summer, and (d) autumn at AMY in 2016

[Figure]

Figure 5. Bivariate polar plots for observed $CO_2$ (L1) in (a) winter, (b) spring, (c) summer, and (d) autumn at JGS in 2016

[Figure]

Figure 6. Bivariate polar plots for observed $CO_2$ (L1) in (a) winter, (b) spring, (c) summer, and (d) autumn at ULD in 2016

7.  P3 L21: "can cool" to "can be cooled"

    Corrected

8.  P3 L21: this sentence might be reformulated to read "…-80° C, which makes the real temperature of inner air flow to be -50° C." The -80° C seems to be a set temperature. Where is the temperature sensor placed?

    The temperature sensor is for the cold trap to keep trap's temperature not for air stream inside that it is located on the cold trap (between the surface and inner chamber). Even though the trap is cooled down to -80 ℃ (the set point), the air that comes from outside through the pump makes temperature inside of chamber increase. If the air doesn't come into the trap from outside, the temperature inside the drying system would reflect the trap temperature as -80 ℃. When monitored the inside temperature with air stream, it was about – 50 ℃. And this is the key technique for this drying system. So we improve the manuscript in section 2.2 to make it clear as correcting according to your comments.

9.  P3 L22: "drops it" and "cools it"—hard to get what "it" means. If I understand correctly, this sentence might be reformulated for instance, "The sample air is cooled to -20° C in the first trap, and then to -50° C in the second trap."

    We corrected according to your comments, please see revised section 2.2.

10. P3 L25: "One of the dual traps is used to dry ambient air for 24 hr while…"

    Corrected

11. P3 L25: here "hr" is used instead of "hours" used at other places.

    Corrected

12. P4 L20: See my comment above. Here the calibration means anchoring the analyzers' responses to the CO2 scale guaranteed by the standards.

    We reconstructed and improved the manuscript in section 2.3.1. Please see the revised manuscript.

13. P4 L22: See my comments above. Here the calibration means to measure working standards against the CO2 scale.

We corrected to make it clear. Please see revised manuscript in section 2.3.1.

14. P4 L22: See my comments above. This sentence is very unclear. Should this be read like "working standards used at AMY are those directly provided by CCL?"

We corrected to make it clear. Please see revised manuscript in section 2.3.1.

15. P4 L23: Insert "the" before "laboratory standards"

Corrected and we re-constructed this section.

16. P4 L23: Is the "laboratory standards" primary standards that realize the WMO scale i.e. those at highest hierarchy at KMA? If this is the case, the term like "KMA primary standards" might better clarify the standard category at KMA. How large $CO_2$ mole fraction range is covered by the "laboratory standards"? And are these "laboratory standards" same as the "4 standard gases" appearing at the beginning of the paragraph? In summary, questions are how many "KMA primary standards" (that cover XX to YY ppm in $CO_2$ mole fraction) are prepared and how many "working standards" (that cover XXX to YYY ppm) are prepared for each station? Please describe these information systematically. Also, with what kind of instrument are the working standards measured against laboratory standards?

We tried to make it clear. "Laboratory standard" is the glossary in WMO/GAW community. And also primary standard normally is considered the standards which are maintained at NOAA (WMO/GAW CCL) within WMO/GAW (http://www.empa.ch/web/s503/gaw_glossary). So we thought it makes readers who are involved in WMO/GAW measurement get confused.
We added the definition for laboratory standard and the references. And also we have 4 laboratory standards at AMY, central lab for Korea GHG network, and 4 working standards at each station. We make it correct. We tried to describe how to propagate the cylinder. Please see reconstructed and revised manuscript in section 2.3.1.

17. P4 L25: "When the scale is propagated" – this is normally called "calibration". I understand that, firstly, working standards are calibrated against "laboratory standards" to assign them the WMO $CO_2$ scale values, and secondly, by analyzing these gases, the instrument responses are linked to the WMO $CO_2$ scale values.

We added the calibration definition according to your comment and try to make it clear.

18. P4 L26: Here "calibration" means to determine the instrument response curve.

    We added the definition and try to make it clear in section 2.3.1

19. P4 L27: Again "calibration" is used same as above.

    We added the definition and try to make it clear in section 2.3.1

20. P4 L30: Please exactly indicate the degree of agreement between KMA and CCL (+/-0.0X±0.0Y ppm) found from the Round Robin.

    Corrected in section 2.3.1

21. P5 L12: "500 ppm". First, I would expect that some CO2-elevated events (caused by "local" or "regional" sources) where CO2 mole fraction exceeds 500 ppm can happen. This data treatment may perhaps lose data with scientific value. Second, since the atmospheric CO2 mole fraction is increasing, I would use a value that well follows the atmospheric trend for instance XX ppm + the long-term trend. The constant value 500 ppm does not mean same as that in past or future years.

    We totally agree with your comments. And we also cannot define all CO2 > 500 ppm are invalid. Since we use the values of 500 ppm for CO2 (due to human issue) and 3000 ppb for CH4 (due to carrier gas from GC) to detect room air due to line leakage, we commented it. So we removed this sentence and also next sentence reflected this issue as well that it is not necessary to comment again.

22. P6 L30: "calibrations"

    Since we corrected section 2.3.1, we let it remain as it is.

23. P7 L24: "1.0±1.9 ppm at ULD"

    Corrected

24. P7 L25: As in my earlier comment, I need to question if the positive values in CO2XS at the stations are simply ascribed to "local activities".

    We improved our manuscript, please see section 3.1.

25. P8 L5: "An automatic weather station…" This sentence might be moved to section 2.1.

   We moved the information to the section 2.1.

26. P8 L15: What are the "tourist activities" specifically? Local transportation?

   We added the information about tourist activities.

27. P8 L23: As in my earlier comment, the wording like signals of Chinese emissions in "baseline" data may be debatable. And "downwind of East Asia" – note that these two stations are also in East Asia.

   We agree with your opinions that we removed the "baseline".
   And we revised "East Asia" as "continental air mass". And we also replaced all words of East Asia to appropriate words within a context.

28. P8 L32: "the degree and speed of atmospheric mixing" If the authors means dynamics of the PBL, they might mention to rectifier effect (e.g. Denning et al. 1999; Chan et al. 2008). The wording might be re-considered. Same comment as P11 L13.

   We improved the wording according to your suggestions in section 3.4

29. P9 L7: Although I am not a non-native in English, I wonder if "plateau" can represents stabilization after decrease (not increase).

   Corrected

30. P9 L29: same comment as to P8 L23. AMY is also in East Asia.

   We revised it as"…transported through the Yellow sea from the Asia Continent…"

31. P9 L10-15: Unfortunately, this paragraph does not try to explain possible causes of the diurnal variation observed in August. The up-valley and down-valley wind feature was already described in section 2.1. Here I hope to see discussions on how such a wind pattern or any possible sources/sinks upwind could affect variations in CO2.

   For three stations, we simply compare the CO2 diurnal cycle, wind directions and temperature not

only in August but also in other seasons (Figure B). The wind pattern is very clear in August at both of AMY and JGS. We could assume that AMY and JGS are affected by land-sea breeze that daytime $CO_2$ is mainly derived from ocean in August. It is very well matched with temperature. After temperature increase or decrease, the wind direction is changed with CO2. And the altitude of those stations is lower compared to ULD, it can be affected by vegetation easily. Therefore those two stations might be affected by the local wind pattern, the boundary layer and the vegetation.

On the other hand, at ULD, the air-mass mainly comes from the sectors between 0 to 90 degree and 210 to 300 regardless of seasons and time, even though the temperature pattern is very similar to other two stations. It means that the wind direction might not be related to the radiative cooling and warming effects. We assumed this is caused by complex terrain with small and high mountains that surround the station. Therefore, at ULD, diurnal CO2 variation is less related to wind pattern but it is affected by active photosynthesis in summer at least. We tried to explain this issue clearly in section 3.4. It was supposed that the valley-mountain wind affected the diurnal pattern of CO2 initially but it is only few cases. Therefore we cannot define that ULD is affected by the wind pattern and it needs to revise the sentences including this comment. We also improved the section 2.1 with detailed geological information around ULD.

(a)

[Figure]

(b)

(c)

Figure B. The wind directions (blue box plot) temperature (orange), and $CO_2$ diurnal cycle (red line) at (a) AMY, (b) JGS, and (c) ULD.

32. P9 L25: "4.8 to 5.8 ppm" and "-6.8 to -9.6 ppm" I guess these values are deviations from a certain value like an annual average from each station. Please explain.

   It is the maximum and minimum values cross the stations that we suggested where the value was derived from on table 3.

33. P10 L15–23: All detailed technical information should be moved to section 2. In this section the authors should focus on how such technical events affected the measurement data.

   We moved the relevant sentences to the technical descriptions to the section 2.2. and 2.3.1 Please confirm the revised Section 2.3.1 Calibration method in the response # 3 as well. We also revised Table 1 regarding to the revised section 2.

Table 1. Information about the three KMA monitoring stations in Korea and the two monitoring stations in East Asia

| Station | ID | Longitude | Latitude | Altitude | Inlet height | Measurement History |
|---|---|---|---|---|---|---|
| Anmyeondo, Korea | AMY | 126.32°E | 36.53°N | 47 m | 20 m | Since 1999 to July, 2004 |
| | | | | | 40 m | Since July, 2004 |
| Jejudo Gosan Suwolbong, Korea | JGS | 126.16°E | 33.30°N | 71.47 m | 6 m | Since 2012 |
| Ulleungdo*, Korea | ULD | 130.90°E | 37.48°N | 220.9 m | 10 m | Since 2012 |
| Mt.Waliguan, China | WLG | 100.90°E | 36.28°N | 3810 m | 5 m | Since 1990 |
| Ryori, Japan | RYO | 141.82°E | 39.03°N | 260 m | 20 m | Since 1987 |

34. P11 L7: "using" to "relative to"

   Corrected

35. P11 L10: "regionally" to "locally"

   Corrected. And also we revised the word "selected CO2 mole fractions" to "The CO2 mole fractions observed .." since "selected" can be understood as "the selected baseline conditions".

36. P11 L11: What is "the long-transported CO2 levels"? High wind speed does not explain the relatively low CO2 level.

   We agree to your opinions and improved sentence in section 4. As you mentioned high wind speed cannot result in long range transported CO2 while it can provide the environment that less

affected by local sources. So we explained it.

37. P11 L15: "Due to its location it is…" My understanding is that the latter "it" means "CO2 mole fraction observed at ULD". If this is the case, this sentence is strange. Mountain and valley breezes cannot change CO2 mole fraction directly.

We agree with your comments and as we described the characteristics of ULD at #31. We removed this sentence since we thought it is very unclear.

38. P11 L17: delete "added"

Corrected

39. Table 3 caption: "abundances" to "mole fractions".

Corrected

40. Table 3: According to the caption, the uncertainties are simple standard deviations calculated from the all data collected during the respective years. It includes signals of the all components: the long-term, seasonal, synoptic-scale and diurnal variation, and results in too big estimates of uncertainties of the annual mean values. Indeed, the numbers tabulated in Table 3 apparently show that there are no significant differences in annual means between every successive years (i.e. no trend is detectable). The authors should calculate uncertainty that better represent an estimate of error of an average.

We agree the referee's comment. The uncertainty is totally different from the standard deviations. On the other hand, to estimate the uncertainty from the measurement is very challenging for not only Korea network but also other stations which we downloaded data from World Data Centre since data provider did not show their uncertainty factors such as repeatability and reproducibility. On the other hand, we explained our measurement uncertainties in section 3.1 to help readers understand our measurement uncertainty, which are different from the hourly, daily and monthly standard deviations. Therefore we just clarified the standard deviation is not uncertainty here as revising the caption of table 3.

Though the referee commented, the similar mean values at ULD are not the result of mis-calculation. We had discussed this before submitting this paper but could not find out clear reason. We just could find that the monthly mean from July to August is lower than those values in previous years. It seems to lead similar annual mean in 2014 to that value in 2013 and low growth

rate in 2014. On the other hand, we cannot distort our data or delete without certain reasons that we just decided to submit the data itself. After taking your comment, we looked at the data again, checked all flags and any other environmental issues, but still unclear. In this time, we addressed this issue directly in the paper for the readers who are interested in it. We added the sentence "
[revised manuscript text omitted]

[Figure]

[Figure]

(a) (b) (c) (d)

Figure 5. Bivariate polar plots for observed $CO_2$ (L1) in winter (a), spring (b), summer (c), and autumn (d) at JGS in 2016

[Figure]

(a) (b) (c) (d) (e) (f) (g) (h)

Figure 6. Bivariate polar plots for (left panel) selected baseline $CO_2$ (L2) and (right panel) polluted $CO_2$ in spring (L1-L2) (a and b), summer (c and d), autumn(e and f), and in winter(g and h) at ULD in 2016.

[Figure]

Figure 6. Bivariate polar plots for observed $CO_2$ (L1) in winter (a), spring (b), summer (c), and autumn (d) at ULD in 2016

(a)

[Figure]

(b)

(c)

Figure 7. Mean diurnal variations of $CO_2$ mole fraction. Values show the average departure from the daily mean in January, April, August and November at (a) AMY, (b) JGS and (c) ULD from 2012 to 2016.

(a)

(b)

[Figure]

Figure 8. The time series of (a) the monthly mean $CO_2$ and (b) the annual growth rate at WLG, AMY, JGS, ULD and RYO. Annual growth rate was defined as the increase in the annual mean of de-seasonal (long term trend) values from the corresponding value in the previous year. The growth rate reported by WMO is overlaid on (b) and this value is annual increase (not de-seasonal), absolute differences from the previous year.

[Figure]

Figure 9. (a) Long-term trend of atmospheric $CO_2$ and its (b) instantaneous growth rate at WLG, AMY and RYO. Overlaid grey line indicated the period of the negative (in 2004 and 2006) and positive (in 2012) growth rates at AMY compared to other two East Asia stations (WLG and RYO).

Table 1. Information about the three KMA monitoring stations in Korea and the two monitoring stations in East Asia

| Station | ID | Longitude | Latitude | Altitude | Inlet height | Measurement History |
|---|---|---|---|---|---|---|
| Anmyeondo, Korea | AMY | 126.32°E | 36.53°N | 47 m | 20 m | Since 1999 to July, 2004 |
| | | | | | 40 m | Since July, 2004 |
| Jejudo Gosan Suwolbong, Korea | JGS | 126.16°E | 33.30°N | 71.47 m | 6 m | Since 2012 |
| Ulleungdo*, Korea | ULD | 130.90°E | 37.48°N | 220.9 m | 10 m | Since 2012 |
| Mt.Waliguan, China | WLG | 100.90°E | 36.28°N | 3810 m | 5 m | Since 1990 |
| Ryori, Japan | RYO | 141.82°E | 39.03°N | 260 m | 20 m | Since 1987 |

*ULD is not GAW station.

Table 2. The uncertainty estimates for measurements of $CO_2$ at each station from 2012 to 2016. Units are ppm. All terms are in the 68% confidence interval

| Uncertainty factors | AMY | JGS | ULD |
|---|---|---|---|
| $U_{h2o}$ | 0.023 | 0.009 | 0.018 |
| $U_p$ | 0.053 | 0.046 | 0.025 |
| $U_r$ | 0.048 | 0.056 | 0.065 |
| $U_{sacle}$ | 0.088 | 0.088 | 0.088 |
| $U_T$ | 0.116 | 0.114 | 0.114 |

Table 3. Annual mean abundances from 2012 to 2016, mean seasonal amplitude and growth rates.  Seasonal amplitudes are calculated from the detrended data. $CO_2$ at ULD, 2012 was calculated only from February to December, without January. Units are dry-air mole fractions (ppm)

| Year | WLG | AMY | JGS | ULD | RYO |
|---|---|---|---|---|---|
| 2012 | 394.7 ± 3.9 | 402.8 ± 3.6 | 399.7 ± 3.7 | 398.4 ± 3.6 | 397.6 ± 3.7 |
| 2013 | 397.2 ± 3.1 | 405.4 ± 4.6 | 402.5 ± 3.5 | 401.8 ± 4.4 | 400.1 ± 4.2 |
| 2014 | 398.6 ± 3.8 | 407.8 ± 5.7 | 403.9 ± 4.0 | 401.9 ± 5.5 | 401.7 ± 5.1 |
| 2015 | 401 ± 3.3 | 410.2 ± 5.7 | 407.0 ± 4.5 | 405.0 ± 5.0 | 404.1 ± 4.4 |
| 2016 | 404.9 ± 3.2 | 412.6 ± 6.1 | 410.0 ± 4.6 | 409.3 ± 5.1 | 407.4 ± 4.5 |
| Mean seasonal amplitude over 5 years. | 12.2 ± 0.9 | 15.4 ± 3.3 | 13.2 ± 1.7 | 14.2 ± 3.1 | 13.5 ± 1.6 |
| *Maximum* | *5.4 ± 0.7* | *5.8 ± 0.7* | *4.8 ± 0.4* | *5.4 ± 1.0* | *5.6 ± 0.4* |
| *Minimum* | *-6.8 ± 0.7* | *-9.6 ± 2.6* | *-8.3 ± 1.3* | *-8.8 ± 2.3* | *-7.9 ± 1.3* |
| Mean annual growth rate over 5 years (ppm·yr$^{-1}$) | 2.5 ± 1.1 | 2.5 ± 0.7 | 2.6 ± 0.9 | 2.5 ± 1.7 | 2.4 ± 0.7 |

---

## Editor Decision (ED1)

Dear Dr Lee,

Thank-you for the work that you and your co-authors have undertaken to address the reviewers comments on your manuscript. Overall I think you have responded sufficiently to the comments from the reviewers. Below, please find a list of technical corrections that I think would improve the clarity of the manuscript in a few places. The line numbers are taken from the author response file with track-changes.

P3, line 37: Is 'air' needed after 'ambient'?

P4, line 5: Reviewer 2 recommended that 'Each trap is employed drying' be replaced with 'One of the dual traps is used to dry ...' but this change doesn't seem to have been made.

P5, line 31: Add 'as' before 'a stepwise change'

P8, line 30: replace 'quote' with 'quoted'

P11, line 18: Suggest 'geographical' rather than 'geological'

P12, line 10: Suggest replacing 'from' with 'by unusually low $CO_2$ in July-August 2014 resulting in'

P13, line 23: Replace '1' with '1.0'

It would be good to add a section at the end of the paper to describe the availability of the data from these sites.

Regards,

Rachel Law, rachel.law@csiro.au

---

## Author Response (AR2)

Authors response to Dr. Law, the Editor. A copy of the reviewer comment is given (with bullet point 'number') followed by a response (blue font).

We very appreciate your acceptance of our paper. We improved our manuscript according to your comments.

1. P3, line 37: Is 'air' needed after 'ambient'?
   Corrected in P3 line 27

2. P4, line 5: Reviewer 2 recommended that 'Each trap is employed drying' be replaced with 'One of the dual traps is used to dry ...' but this change doesn't seem to have been made.
   Corrected in P3 line 29

3. P5, line 31: Add 'as' before 'a stepwise change'
   Corrected in P5 line 16

4. P8, line 30: replace 'quote' with 'quoted'
   Corrected in P7 line 35

5. P11, line 18: Suggest 'geographical' rather than 'geological'
   Corrected in P10 line 6

6. P12, line 10: Suggest replacing 'from' with 'by unusually low CO2 in July-August 2014 resulting in'
   Corrected in P10 line 30

7. P13, line 23: Replace '1' with '1.0'
   Corrected in P11 line 35

8. It would be good to add a section at the end of the paper to describe the availability of the data from these sites.
   We added the section 2.3.4 Data availability. We now provide our L2 data to the public through the World Data Centre for Greenhouse Gases (http://gaw.kishou.go.jp) only for AMY and JGS. However we have a plan to open ULD data in the near future, we mentioned it in a new section of 2.3.4 in P6 line 13
   One thing we worried is the monthly mean in WDCGG is calculated with different method from the method by Thoning et al.(1989) in section 2.3.3. We were supposed to explain this method, how to calculate the monthly data in WDCGG, in the paper but since it could be diffuse as increasing of confusions, we did not describe it. Instead, we replace the "Finally" with "In this paper" in section 2.3.3 in P6 line 9.

9. It was not related to Editor's comment but our funding title and the number were changed that we also revised lines in our "Acknowledgement".

[revised manuscript text omitted]